

# Superimposed vibration on suspended push-ups

Bernat Buscà[1], Joan Aguilera-Castells[1], Jordi Arboix-Alió[1,2], Adrià Miró[1], Azahara Fort-Vanmeerhaeghe[1,2], Pol Huertas[1] and Javier Peña[3,4]

[1] Faculty of Psychology, Education Sciences, and Sport Blanquerna, Ramon Llull University, Barcelona, Spain
[2] School of Health Science Blanquerna, Ramon Llull University, Barcelona, Spain
[3] Sport and Physyical Activity Studies Centre (CEEAF), University of Vic-Central University of Catalonia, Vic, Spain
[4] Sport performance Analysis Research Group (SPARG), University of Vic-Central University of Catalonia, Vic, Spain

Corresponding author
Joan Aguilera-Castells,
joanac1@blanquerna.url.edu

## ABSTRACT

**Background.** Superimposition of vibration has been proposed in sports training using several devices and methods to enhance muscle activation and strength adaptations. Due to the popularity of suspension training, vibration systems have recently been developed to increase the effects of this training method. The present cross-sectional study aims to examine the effects of superimposing vibration on one of the most popular exercises in strength and conditioning programs: push-ups.

**Methods.** Twenty-eight physically active men and women executed push-ups in three suspended conditions (non-vibration, vibration at 25 Hz, and vibration at 40 Hz). OMNI-Res scale was registered, and surface electromyographic signals were measured for the activity of the right and left external oblique, anterior deltoid, triceps brachii, sternal, and clavicular heads of the pectoralis major.

**Results.** A linear mixed model indicated a significant fixed effect for vibration at 25 Hz and 40 Hz on muscle activity. Suspended push-ups with superimposed vibration (25 Hz and 40 Hz) showed a significant higher activity on left (25 Hz: $p = 0.036$, $d = 0.34$; 40 Hz: $p = 0.003$, $d = 0.48$) and right external oblique (25 Hz: $p = 0.004$, $d = 0.36$; 40 Hz: $p = 0.000$, $d = 0.59$), anterior deltoid (25 Hz: $p = 0.032$, $d = 0.44$; 40 Hz: $p = 0.003$, $d = 0.64$), and global activity (25 Hz: $p = 0.000$, $d = 0.55$; 40 Hz: $p = 0.000$, $d = 0.83$) compared to non-vibration condition. Moreover, OMNI-Res significant differences were found at 25 Hz ($6.04 \pm 0.32$, $p = 0.000$ $d = 4.03$ CI $= 3.27, 4.79$) and 40 Hz ($6.21 \pm 0.36$ $p = 0.00$ $d = 4.29$ CI $= 3.49, 5.08$) compared to the non-vibration condition ($4.75 \pm 0.32$).

**Conclusion.** Superimposing vibration is a feasible strategy to enhance the muscle activity of suspended push-ups.

# INTRODUCTION

Combining different strength training methods is an increasingly used strategy to reach sports performance and competitive advantages. The synergistic effect of recruiting prime movers, antagonists, and stabilizers, justifies the use of complex exercises that present

instability (*La Scala Teixeira et al., 2019*). This effect can be even more important in sports, where perturbed tasks constitute the essence of their specificity (*Behm & Anderson, 2006*). The upper body muscles can benefit from instability, especially in overhead disciplines, such as handball, water polo or hockey, and gymnastic sports, continuously demanding precise, powerful, complex, and unidirectional actions. Acting as a pendulum by rotating around a singular anchor point above, suspension training uses its essential characteristics (vector resistance, stability, and pendulum) and body weight to enhance neuromuscular demands (*Bettendorf, 2010*).

Complex tasks involving instability have been combined with mechanical vibrations to increase its neuromuscular demands in the past (*Cloak et al., 2013*; *Marín & Hazell, 2014*; *Ritzmann et al., 2014*; *Sierra-Guzmán et al., 2018*; *Aguilera-Castells et al., 2019*; *Aguilera-Castells et al., 2021*). Vibratory training transfers vibration on the muscle to elicit the tonic vibration reflex (*Cardinale & Bosco, 2003*). Superimposing vibration can alter the motor unit recruitment, activating faster and larger motor units (*Martin & Park, 1997*; *Xu et al., 2018*), thus reinforcing the possible benefits of using those devices in standard training methods (*Cardinale & Wakeling, 2005*). Whole-body vibration (WBV) applied through platforms is the most studied vibrating method to provoke acute neuromuscular effects (*Cardinale & Lim, 2003*; *Rønnestad, 2009a*; *Rønnestad, 2009b*) and long-term adaptations (*Gollhofer, 2010*; *Manimmanakorn et al., 2014*). In contrast, other studies demonstrated no significant chronic effects of vibration in parallel squat 1RM (*Rønnestad et al., 2012*) and elbow flexion 1RM (*Drummond et al., 2014*). Nevertheless, several devices superimposed vibration on barbells (*Poston et al., 2007*; *Mischi & Cardinale, 2009*; *Moras et al., 2010*; *Xu, Rabotti & Mischi, 2013*), dumbbells (*Bosco, Cardinale & Tsarpela, 1999*; *Cochrane & Hawke, 2007*), and cables (*Issurin & Tenenbaum, 1999*; *Issurin et al., 2010*) have also been designed to transfer vibratory stimuli to the upper body. *Mischi & Cardinale (2009)* reported significantly higher muscle activity in arm muscles when performing the isometric V exercise. *Moras et al. (2010)* have demonstrated the acute effects of superimposing vibration in a bench press in the prime movers, especially during flexion. *Poston et al. (2007)* showed a greater bench press average power in a vibrating condition, although they did not assess muscle activity. The authors superimposed the vibrating engine on the barbell side. Similarly, *Xu, Rabotti & Mischi (2015)* prototyped a bench to combine the effect of muscle tension and vibration on muscle activation, demonstrating the benefits of using an adaptive normalized least mean square algorithm to determine the real effects of superimposed vibration on the biceps brachii. Lately, vibration has been superimposed on a suspension device in lower limb exercises (*Aguilera-Castells et al., 2021*). When performing dynamic supine bridges and hamstring curls, surface electromyography reflected a higher activity of the muscles proximal to the straps (gastrocnemius medialis, lateralis, and semitendinosus). However, the effect on the primary movers was non-significant.

Beyond the physiological markers, reporting the perception of how hard the load is lifted has been a recurrent method to guide and monitor the strength training programs. For this purpose, OMNI Perceived Exertion Scale for Resistance Exercise (OMNI-Res) has been developed by *Robertson et al. (2003)* and validated in several strength and conditioning contexts with promising results (*Robertson et al., 2005*; *Lagally & Robertson, 2006*; *Colado*

*et al., 2012*; *Bautista et al., 2014*; *Buscà et al., 2020*). Indeed, the mentioned studies reported that the scale is strongly connected to 1RM, muscle activity, or total weight lifted in different training environments. Furthermore, other studies have demonstrated the increased exertion perception performing the exercises under vibration conditions (*Marín et al., 2012a*; *Marín et al., 2012b*; *Aguilera-Castells et al., 2021*).

Therefore, the main objective of the present study was to examine the effects of vibration on muscle activity in dynamic suspended push-ups. It was hypothesized that the superimposed vibration on the suspension straps might obtain higher muscle activity than the suspended condition without vibration. It was also hypothesized that the OMNI-Res perceived exertion scale for resistance exercise would be higher in the vibration exercises than the non-vibrating exercises.

## MATERIALS & METHODS

### Design
A cross-sectional study design investigated the effects of a suspension device with superimposed vibration on upper body muscle activity. Participants performed suspended push-ups in non-vibration, vibration at 25 Hz and 40 Hz. Surface electromyography (sEMG) was used to record and compare the activity of the right and left external oblique, anterior deltoid, triceps brachii, and sternal and clavicular heads of the pectoralis major. sEMG values were expressed as a percentage of maximum voluntary isometric contraction (% MVIC). Furthermore, the perceived exertion was assessed using the OMNI-Perceived Exertion Scale for Resistance Exercise (OMNI-Res) under all suspended push-ups conditions (Fig. 1).

### Participants
Twenty-eight physically active male ($n = 25$, mean age = $22.7 \pm 3.6$ years, height = $1.8 \pm 0.1$ m, body mass = $77.7 \pm 8.4$ kg, body mass index = $24.5 \pm 2.1$ kg m$^{-2}$, suspension training experience = $5.2 \pm 2.7$ years) and female ($n = 3$, mean age = $22.6 \pm 0.6$ years, height = $1.6 \pm 0.0$ m, body mass = $56.0 \pm 4.0$ kg, body mass index = $21.9 \pm 2.4$ kg m$^{-2}$, suspension training experience = $3.7 \pm 2.5$ years) voluntarily participated in the study. Participants were excluded from participating in the study if their suspension training experience was under one year, did not perform a minimum of 90 min of physical activity per week, or had cardiovascular, musculoskeletal, or neurological diseases. Before beginning the familiarization session, all participants were informed of all study procedures, benefits, and risks, in oral and written form, before receiving and signing the informed consent form. The Physical Activity Readiness Questionnaire (PAR-Q) was also handed out to the participants to identify any health risks related to physical exercise (*Warburton et al., 2011*). Three to four hours before the testing sessions participants did not ingest any stimulant substances (e.g., caffeine), food or drink. In addition, no high-intensity physical activity was performed 24 h before the tests. The Ethics and Research Committee Board of Blanquerna Faculty of Psychology and Educational and Sport Sciences at Ramon Llull University in Barcelona, Spain (ref. number 1819005D) approved this study, and the protocols followed the principles of the Declaration of Helsinki (revised in Fortaleza, Brazil, 2013).
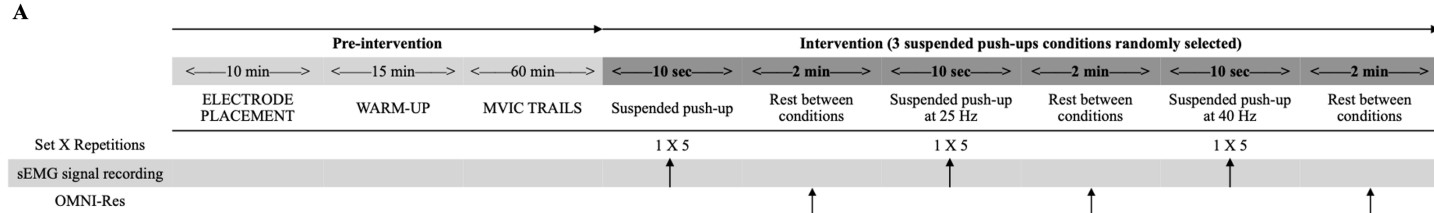

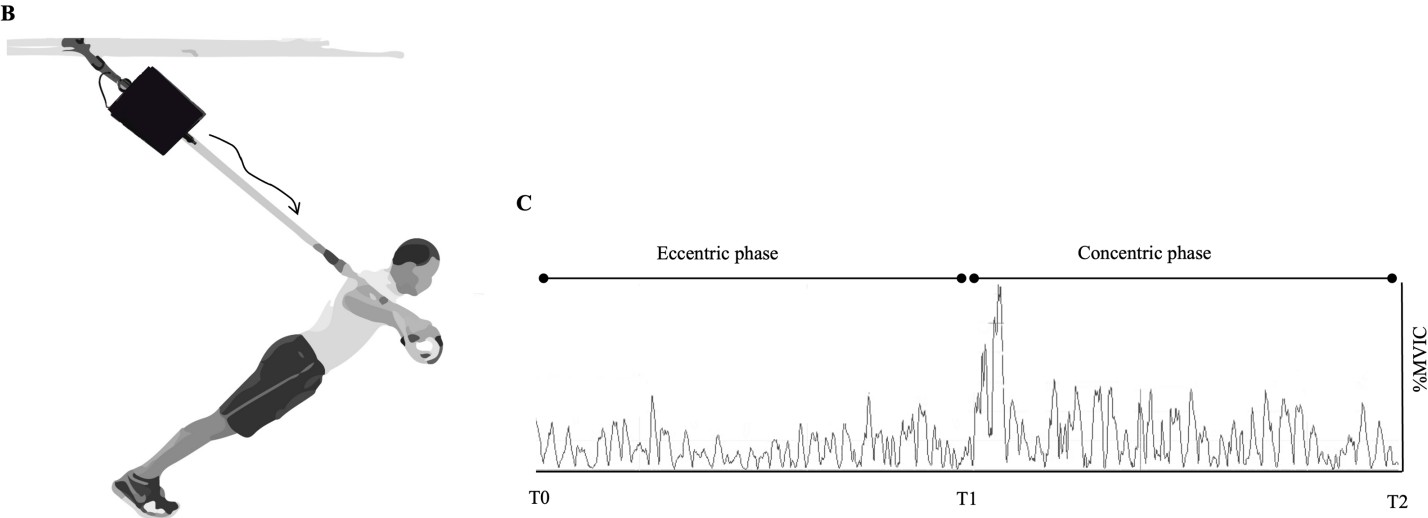

**Figure 1** **Study design, standardized suspended push-up, and sEMG signal.** (A) Schematic representation of the study design with the timeline of the different procedures. The vertical arrow indicates the recording of the sEMG signal and the collection of the OMNI-Res. (B) Suspended push-up under vibration conditions. The black box represents the vibration device. The sketched arrow indicates the direction and transmission of vibration through the suspension straps. (C) Sample of the sEMG signal of the clavicular head of pectoralis major during a repetition of the suspended push-up. The eccentric phase is the range of motion from the upper to the lower limit, and the concentric phase from the lower to the upper. T0 = start of the suspended push-up; T1 = end of the eccentric phase and onset of the concentric phase; T2 = end of the suspended push-up.

## Procedures

A familiarization session was held one week before the testing sessions. Participants were instructed to perform the suspended push-ups with proper technique in the different conditions (non-vibration, vibration at 25 Hz and 40 Hz) in two sets of five repetitions. Anthropometric (*e.g.*, weight, height, acromion distance) and training experience data were collected. The test session was carried out a week later and at the same time in the morning. Researchers cleaned the electrode site with alcohol, shaving the skin area when necessary, thus placing the surface electrodes (Biopac EL504 disposable Ag-AgCl; BIOPAC System, Inc., Goleta, CA) on the external oblique (left and right), anterior deltoid, triceps brachii, sternal and clavicular portion of pectoralis major on the dominant upper limb (*Criswell & Cram, 2011*). A reference electrode over the iliac crest was placed and all electrodes were placed at an inter-electrode distance of two cm following the SENIAM guidelines (*Hermens et al., 2000*). Next, a standardized warm-up consisting of 10 min of dynamic upper body calisthenics and two sets of eight repetitions of strict push-ups on the floor was performed. Then, participants executed a maximal voluntary isometric contraction (MVIC) test for the right and left external oblique, anterior deltoid, triceps brachii, and the

sternal and clavicular head of the pectoralis major. The MVIC values were used as a baseline to normalize the sEMG signal (*Halaki & Ginn, 2012*). After the normalization protocols, participants completed a set of five dynamic repetitions for each push-up condition in a randomized order. The standardized suspended push-up technique consisted of holding the legs at shoulder-width apart, the hands separated at 150% of the acromial distance, in a pronated position, and grabbing the suspension strap handles (TRX Suspension Trainer; Fitness Anywhere, San Francisco, CA, USA). Throughout the exercise, participants were instructed to maintain the lower back natural sway. For the lower position during the suspended push-up conditions, the elbow flexion was standardized at 90° and measured using a goniometer. Customized stoppers (similar to hurdles) were used to control the elbow flexion and 150% acromial width. Participants began the suspended push-up in the upper position (elbow extension) with a plantar flexion over the plumb line between the anchor point of the suspension strap and the ground. Participants flexed their elbows to 90° (lower position) in this position, then pushed with their hands on the handles to extend their elbows and return to the upper position (Fig. 2). The length of the suspension strap was standardized at 1.64 m, and the inclination ranged from 20° to 33° (mean ± SD: 26.5° ± 3.5). A positional transducer (WSB 16k-200; ASM Inc., Moosinning, DEU) was used to control the range of movement in each suspended push-up condition. The positional transducer tether was attached to the chest. The measured signal was used to identify the beginning and end of each repetition and determine the eccentric phase (lower position) and the concentric phase (upper position) of the suspended push-up. The pace of the push-up repetitions was standardized using a metronome settled at 60 beats per minute (1 s per phase). Furthermore, all participants were given two-minute rest for each suspended push-up attempt. Those repetitions that did not follow the standard technique established by the researchers were repeated with two-minute rest between attempts. All participants were asked about the possible discomfort from the vibration exposure on the head or other body regions. None of them reported discomfort.

A vibration device provided the superimposed vibration on the suspension straps for suspension training settled at two frequencies (25 Hz and 40 Hz) with an amplitude of eight mm (peak to peak). The device was attached between the ceiling anchor point and the suspension strap to transmit the vibration through the straps using a connecting rod's vertical motion caused by an electric motor's rotary motion.

## Muscle activity assessment

Muscle activity of the analyzed muscles during suspended push-ups (non-vibration, vibration at 25 Hz and 40 Hz) was obtained using the six-channel sEMG BIOPAC MP-150 System (sampling rate: 1.0 kHz; BIOPAC System, Inc., Goleta, CA). The sEMG was processed using a bandpass filtered at 10–500 Hz with a fourth-order 50 Hz Butterworth notch filter. The motion artifacts were remove using additional notch filters and applied for the 25 Hz and 40 Hz vibration frequencies, as recommended *Borges et al. (2017)*. Then, the root mean square (RMS) algorithm with a moving window of 150 ms with 50 ms overlap was used to smooth the sEMG signal. Afterward, the sEMG signal was normalized using the maximum smoothed sEMG activity reached by each muscle group in the different MVIC

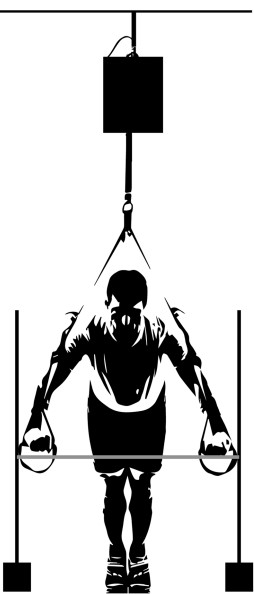

**Figure 2 Suspended push-up: frontal view.** Detail of the customized stops (similar to hurdles) used to standardize the elbow flexion and the acromial width. The black box represents the vibration device.

trails. The normalized sEMG signal was expressed as a percentage of the MVIC (% MVIC). This process was done with the AcqKnowledge 4.2 software (BIOPAC System, Inc., Goleta, CA). In order to normalized the sEMG signal, the MVIC protocol consisted of performing an MVIC for 5 s, increasing the contraction progressively for 2 s and maintaining the MVIC for 3 s, with a 3-minute rest between attempts. The best of the three 5-second attempts was used to normalize the EMG signal (*Jakobsen et al., 2013*). The different positions to reach the MVIC followed the *Konrad (2006)* guidelines, thus for the sternal and clavicular head of pectoralis major the participants lay supine, with their feet on the floor and pushed with their arms (elbows 90°) against an immovable resistance (fixed bar); for the anterior deltoid, the participants sat on a bench with their feet on the floor and leaned their back against the backrest to perform a glenohumeral flexion movement by holding a fixed bar with the hand closed in a pronated position blocking the elbows; in the previous position, the bar was adjusted to allow the participants to hold their elbows at 90°, and in this position to perform an extension movement of the elbows against the fixed bar to perform the MVIC of the triceps brachii; for the external oblique (left and right) participants laid on a bench in a side position with their legs and hips held with ratchets, in this position a manual resistance was applied against the lateral trunk flexion movement.

## Perceived Subjective Exertion Measurement (OMNI-Res scale)

After performing each suspended push-up condition, participants were asked about perceived subjective exertion using the OMNI-Res scale and following *Robertson et al. (2003)* protocol. During the familiarization session, a visual OMNI-Res scale was displayed to ensure that participants provided an accurate perception of the exertion. Participants were asked to report their subjective perception of exertion values ranging from 0 (extremely

easy) to 10 (extremely hard). Participants were instructed that on the OMNI-Res scale, a value of 0 is equivalent to performing an unweighted exercise and a value of 10 is equivalent to lifting one repetition maximum. In the test session, the protocol mentioned above was followed, and the OMNI-Res value of each exercise condition was recorded.

## Data analysis

The data analysis consisted of determining the differences in each analyzed muscle's peak activation (%MVIC) in the different conditions of the suspended exercise (push-ups non-vibration, 25 Hz, and 40 Hz) for three intermediate repetitions. Thus, each suspended push-up condition's first and fifth repetition were discarded. In addition, the peak sEMG was analyzed for the concentric (upper position) and eccentric (lower position) phases of the three repetitions. For OMNI-Res, all recorded values after each exercise condition were analyzed as mean OMNI-Res.

## Statistical analyses

The power analysis and the sample size were calculated with the General Linear Mixed Model Power and Sample Size software (GLIMMPSE; version 3.0.0) (*Kreidler et al., 2013*). For a sample of 28 participants, GLIMMPSE showed power of 0.95 and $\alpha$ level of 0.05. The Shapiro–Wilk test was used to determine if dependent variables were normally distributed, except the OMNI-Res. The dependent variables were: (i) the peak EMG amplitude (right and left external oblique, anterior deltoid, triceps brachii, sternal and clavicular heads of the pectoralis major), (ii) mean value in these muscles (global activity), and (iii) the OMNI-Res values. Data from all dependent variables were shown as mean $\pm$ standard error of the mean (SE). An inferential parametric test, a linear mixed model, was carried out to determine the acute effects of suspended push-ups conditions (non-vibration, 25 Hz and 40 Hz) on each analyzed muscle and the global activity. The linear mixed model used muscle activity and global activity as dependent variables, suspended push-ups conditions as the fixed effect, and participants as the random effect. If the linear mixed model showed a statistically significant fixed effect ($p < 0.05$), post hoc comparisons with Bonferroni correction were conducted.

For the previous model, the significance of the fixed effects associated with the outcome variable included in the model was assessed using the Wald test, with statistical significance set at $p < 0.05$. After the models were validated, the residuals of the final models were explored for normality, homogeneity, and independence assumptions. The normality assumption of the residuals was checked using a normal Q–Q plot of residuals.

The effect of suspended push-ups conditions on OMNI-Res was established using a non-parametric Friedman test. A post hoc Wilcoxon test with the Bonferroni correction was carried out in case of a significant main effect. *Cohen*'s (*1988*) *d* effect size with 90% confidence intervals (CI) were calculated and interpreted as trivial (<0.2), small (from 0.2 to 0.6), moderate (from 0.6 to 1.2), large (from 1.2 to 2.0), and very large (>2.0) (*Hopkins et al., 2009*). The SPSS statistical software (version 26; SPSS Inc., Chicago, IL, USA) was used to conduct the statistical data analyses setting the *p*-value at <0.05.

## RESULTS

The normalized sEMG (% MVIC) values for each analyzed muscles under suspended push-ups conditions for concentric and eccentric phase are shown in Table 1, with the fixed effect of the exercise condition on muscle activity. For the concentric phase, the suspended push-ups with superimposed vibration (25 Hz and 40 Hz) showed a significant higher activity on left (25 Hz: $p = 0.036$, $d = 0.34$; 40 Hz: $p = 0.003$, $d = 0.48$) and right external oblique (25 Hz: $p = 0.004$, $d = 0.36$; 40 Hz: $p = 0.000$, $d = 0.59$), anterior deltoid (25 Hz: $p = 0.032$, $d = 0.44$; 40 Hz: $p = 0.003$, $d = 0.64$) and global activity (25 Hz: $p = 0.000$, $d = 0.55$; 40 Hz: $p = 0.000$, $d = 0.83$) compared to non-vibration condition. Superimposed vibration at 25 Hz on the suspension strap provoked a significant small increase on the sternal head of pectoralis major compared to non-vibration condition ($p = 0.007$, $d = 0.39$). For triceps brachii and clavicular head of pectoralis major a significant small increase on activity was found under suspended push-up at 40 Hz compared to non-vibration condition ($p = 0.007$ $d = 0.47$, $p = 0.000$ $d = 0.60$; respectively). Moreover, the standardized differences at 90% CI for the suspended push-ups conditions are represented as forest plots (Figs. 3 and 4). For the eccentric phase, superimposed vibration (25 Hz and 40 Hz) significantly increased left (25 Hz: $p = 0.034$, $d = 0.41$; 40 Hz: $p = 0.002$, $d = 0.53$) and right external oblique (25 Hz: $p = 0.024$, $d = 0.33$; 40 Hz: $p = 0.000$, $d = 0.64$), and the sternal head of pectoralis major activity (25 Hz: $p = 0.013$, $d = 0.35$; 40 Hz: $p = 0.000$, $d = 0.51$) compared to suspended push-up without vibration. Additionally, a significant small increase of right external oblique activity was found under suspended push-up at 40 Hz, in comparison with superimposed vibration at 25 Hz ($p = 0.035$, $d = 0.29$).

Figure 5 shows the OMNI-Res comparison under suspended push-up conditions. A significant main effect was found on suspended push-up conditions on OMNI-Res [$X^2(2) = 26.805$ $p = 0.000$]. The perceived subjective exertion (OMNI-Res) was significantly higher for suspended push-ups at 25 Hz ($6.04 \pm 0.32$, $p = 0.000$ $d = 4.03$ CI = 3.27, 4.79) and 40 Hz ($6.21 \pm 0.36$, $p = 0.000$ $d = 4.29$ CI = 3.49, 5.08) compared to non-vibration ($4.75 \pm 0.32$). A non-significant difference on OMNI-Res was found between suspended push-ups at 25 Hz and 40 Hz ($p = 0.867$, $d = 0.50$ CI = 0.05, 0.94).

## DISCUSSION

The present study showed that superimposing vibration to an upper limb suspended push-up is beneficial, increasing the analyzed muscles' global activity. This primary finding reinforces the evidence that combining different strength methods can elicit superior muscular demands (*Poston et al., 2007*; *Mischi & Cardinale, 2009*; *Moras et al., 2010*; *Xu, Rabotti & Mischi, 2013*). These results can be relevant for coaches and practitioners trying to optimize the time spent in strength and conditioning practices, especially in team-sport settings where time devoted to sessions is limited in-season.

Push-up is one of the most used upper body exercises in sports training. Moreover, it has become even more popular since several authors (*Snarr et al., 2013*; *Calatayud et al., 2014a*) have demonstrated that push-ups can be more challenging under suspension or using unstable environments (*Calatayud et al., 2014b*; *De Araújo et al., 2020*). In the

**Table 1  sEMG activity for each analyzed muscle under suspended push-up conditions.**

| Muscles[a] | Suspended push-up | | | | |
| --- | --- | --- | --- | --- | --- |
| | Non-Vibration Mean ± SE | Vibration at 25 Hz Mean ± SE | Vibration at 40 Hz Mean ± SE | F | p |
| **Concentric phase** | | | | | |
| External oblique_R | 4.30 ± 0.40 | 5.08 ± 0.43[*] | 5.60 ± 0.44[*] | 15.81 | 0.000 |
| External oblique_L | 4.08 ± 0.40 | 4.82 ± 0.44[*] | 5.09 ± 0.40[*] | 6.67 | 0.003 |
| Triceps brachii | 29.04 ± 1.18 | 30.26 ± 1.71 | 33.36 ± 2.14[*] | 5.46 | 0.007 |
| Anterior deltoid | 41.18 ± 2.10 | 47.05 ± 2.86[*] | 48.84 ± 2.39[*] | 6.51 | 0.003 |
| Pectoralis major_S | 24.81 ± 2.02 | 29.23 ± 2.27[*] | 27.77 ± 2.20 | 5.32 | 0.008 |
| Pectoralis major_C | 38.67 ± 2.47 | 42.26 ± 2.44 | 46.16 ± 2.28[*] | 9.99 | 0.000 |
| Global activity[b] | 23.68 ± 0.85 | 26.45 ± 1.05[*] | 27.80 ± 1.02[*] | 24.15 | 0.000 |
| **Eccentric phase** | | | | | |
| External oblique_R | 4.46 ± 0.35 | 5.11 ± 0.40[*] | 5.72 ± 0.40[**] | 14.34 | 0.000 |
| External oblique_L | 4.18 ± 0.34 | 4.96 ± 0.37[*] | 5.27 ± 0.43[*] | 7.17 | 0.002 |
| Triceps brachii | 26.55 ± 1.48 | 28.54 ± 1.81 | 27.44 ± 1.83 | 1.10 | 0.337 |
| Anterior deltoid | 43.22 ± 2.64 | 42.13 ± 3.08 | 39.78 ± 3.37 | 1.35 | 0.266 |
| Pectoralis major_S | 19.08 ± 1.26 | 22.09 ± 1.92[*] | 23.42 ± 1.90[*] | 9.73 | 0.000 |
| Pectoralis major_C | 31.50 ± 3.09 | 32.62 ± 3.22 | 35.10 ± 3.01 | 2.54 | 0.087 |
| Global activity[b] | 21.50 ± 1.01 | 22.57 ± 1.21 | 22.79 ± 1.18 | 2.84 | 0.066 |

**Notes.**
[*]Significantly different with non-vibration condition.
[**]Significantly different with vibration at 25 Hz condition.
[a]Data presented as normalized muscle activity (%MVIC).
[b]Global activity = mean of the six muscles.
C, clavicular head; L, left; R, right; SE, standard error of the mean; S, sternal head.

present study, no comparison was performed between traditional and suspended push-ups; nevertheless, it was hypothesized that the combination of vibration and suspension increases the activation of the primary push-up movers and, probably, the stabilizers of the action. The hypothesis was mainly confirmed in all the analyzed muscles, especially in the concentric phase at 40 Hz. Several authors have demonstrated differentiated effects depending on vibration frequencies, mainly in lower body muscles (*Hazell, Jakobi & Kenno, 2007*; *Di Giminiani et al., 2013*). However, the effects on the upper body muscles are still unclear. In this vein, the addition of vibration at a higher rate of frequencies tested (30–40 Hz) has been shown as an activity enhancer in this and similar exercises performed in a vibration plate (*Ashnagar et al., 2016*; *Grant et al., 2019*). Thus, superimposing vibration seems to be a proper strategy to enhance muscle activity in a suspended push-up. However, no significant differences were found between 25 Hz and 40 Hz when overall muscle activity was considered.

The anterior deltoid is the most differentially demanded muscle under superimposed vibration (*Grant et al., 2019*) or oscillatory vibration exercises (*Arora et al., 2013*). Its role as a prime mover of shoulder adduction and stabilizer of the shoulder joint, together with the proximity to the vertical plane of the vibration transmission, might explain this finding (*Aguilera-Castells et al., 2021*). Furthermore, body inclination (20° to 33°), type of grip, and angle between the straps and the floor reinforce the role of the anterior deltoid

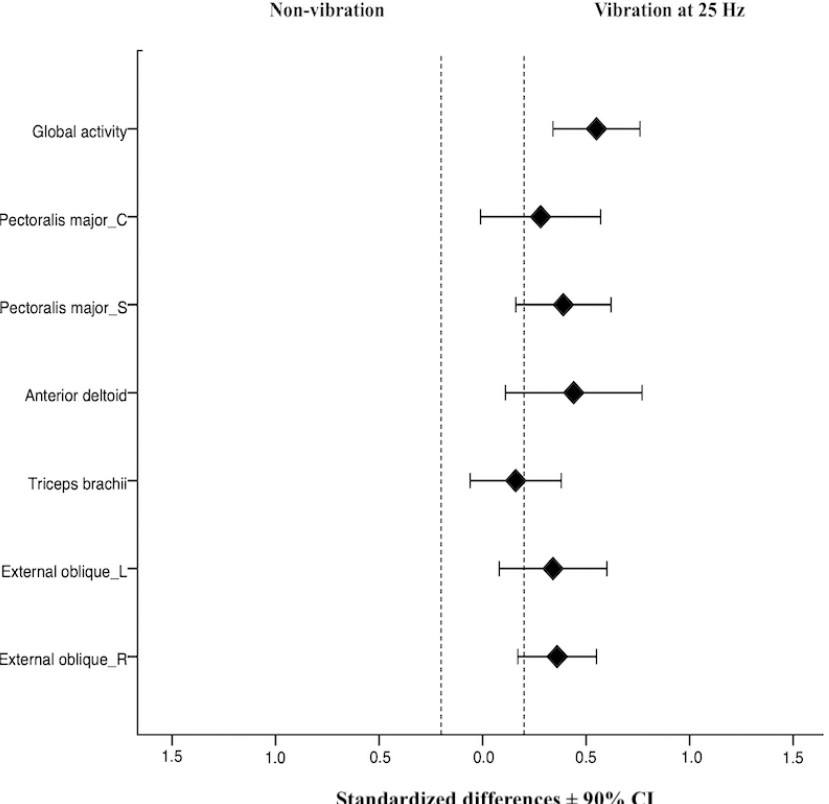

**Figure 3** **Acute effects of superimposed vibration at 25 Hz and suspended push-up without vibration on sEMG activity for each analyzed muscle at concentric phase.** Bars represent the 90% confidence interval for the effect of superimposed vibration at 25 Hz on suspended push-up. Dotted lines represent the smallest substantial threshold. C, clavicular head; Global activity, mean of the six muscles; L, left; R, right; S, sternal head.

in stabilizing the shoulder. However, this muscle is less active under unstable conditions (*Calatayud et al., 2014b*; *Borreani et al., 2015*; *Youdas et al., 2020*). As contributors to arm flexo-extension and shoulder adduction, these two muscle heads are close to the vertical plane and proximal to vertical vibration transmitted through the suspension strap. The activity of the anterior deltoid is probably not enough to dampen vibration. Both heads of the pectoralis major have the additional work to perform and stabilize the suspended dynamic push-up, especially the clavicular head at 40 Hz (Fig. 4) closer to the vibration point. Furthermore, the technique used in this study, with straps situated inside the grip, could explain the present findings. Indeed, this type of grip, with less distance between the two handles, makes the action more unstable, and the main involved muscles are overstimulated by the effect of vibration (*Aguilera-Castells et al., 2019*). This might also be the case with triceps brachii at 40 Hz. If superimposed vibration improves the quality of the strength exercises that recruit this muscle by raising the muscle activation, this effect

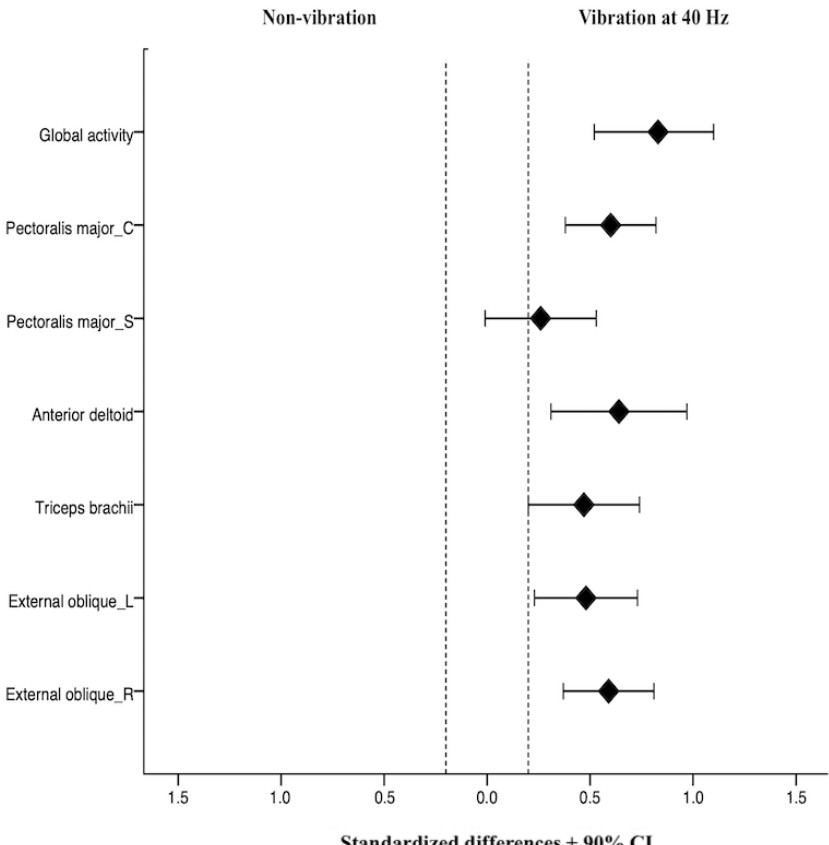

**Figure 4   Acute effects of superimposed vibration at 40 Hz and suspended push-up without vibration on sEMG activity of each analyzed muscle at the concentric phase.** Bars represent the 90% confidence interval for the effect of superimposed vibration at 40 Hz on the suspended push-up. The dotted lines represent the smallest substantial threshold. C, clavicular head; Global activity, mean of the six muscles; L, left; R, right; S, sternal head.

could potentially help to reduce injuries in overhead athletes. Deltoids are synergists of the rotator cuff muscles; these muscles are typically torn by overuse when athletes present shoulder impingement (*Page, 2011*), one of the most frequent injuries in these disciplines.

Although the action plane of this muscle during most of the range of movement is not vertical, the action of the triceps brachii before the complete extension of the arm at the end of the concentric phase is aligned with the vertical transmission of vibration and thus overstimulated. The effect is even more apparent in the adducted technique proposed in the present study (*Cogley et al., 2005*). This conclusion agrees with *Moras et al. (2010)* comparing the vibration effects of pushing a vibratory bar and *Mischi & Cardinale (2009)* pushing an electromagnetic arm actuator involving biceps and triceps brachii. Both studies used frequencies around 30 Hz.

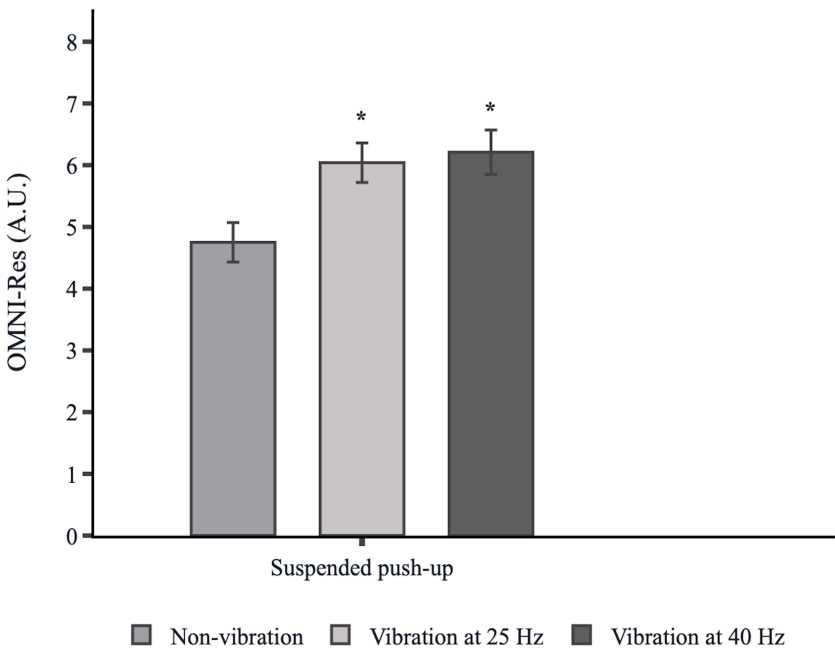

**Figure 5** **Perceived subjective exertion under suspended push-up conditions.** Bars represent the mean of OMNI-Res values, and the error bars represent the standard error of the mean (SE). A.U., arbitrary units. An asterisk (*) indicates that it is significantly different from the non-vibration condition.

According to *Chen et al. (2019)*, in the present study, both external obliques were significantly higher stimulated with the superimposition of vibration. The external oblique is located on the lateral and anterior parts of the abdomen. It is a broad, thin, and irregularly quadrilateral muscle whose muscular portion holds the side. Its aponeurosis is in the anterior wall of the abdomen, and the anterior internal oblique is deep below the anterior external oblique. Thus, in contrast to other studies where no effects were found for the most distal muscles from the vibration exposure point (*Aguilera-Castells et al., 2021*), the present findings evidenced the increased activation of abdominals (*Wirth, Zurfluh & Müller, 2011*). The superior fatigue of the core muscles induced by the vibration, especially in suspended exercises, might be a relevant factor (*Behm et al., 2010*; *Mok et al., 2015*). Indeed, core muscles need to use more energy to maintain posture in an unstable environment, and muscle activity increases (*Cuğ et al., 2012*; *Panza et al., 2014*). Again, the role of core muscles in athletic performance and injury prevention it is not negligible (*Cissik, 2011*). Higher activation of these muscles by means of superimposed vibration may have a superior protective effect in athletes at risk.

OMNI-Res results showed a significant increase in effort perception in both vibration frequencies for the non-vibration condition. Similarly, *Marín et al. (2012b)* found significant increased RPE when performing a squat + biceps curl on a WBV platform and *Aguilera-Castells et al. (2021)* in suspended supine bridge and hamstring curl superimposing vibration. The authors found significant OMNI-Res increases in all

vibration conditions in this work. This finding suggests that the superimposition of vibration is always perceived as a more demanding condition (*Marín et al., 2012a*).

The present study investigated a limited variety of vibration frequencies. Although the used frequencies are the most studied, lower to 25 Hz and higher than 40 Hz should be considered for future research. In addition, the number of sEMG channels limited the number of muscles analyzed. Thus, one could have observed the role played by other allegedly secondary muscles during the different phases of the push-up exercise. Indeed, the cocontraction phenomenon between agonists and antagonists of an unstable upper limb task (*Behm & Anderson, 2006*) can be explored when adding the additional stimulus provided by the vibratory system (*Rodríguez Jiménez et al., 2015*). Since the exercise was dynamic, the wired system used for the sEMG assessment and its compatibility with the suspended push-up forced the investigators to choose a proper technique for avoiding electrode removals. For this reason, the effects of vibration found in the studied push-up technique might not be generalized to other types of execution. The present study was conducted with trained individuals. All of them experienced in suspension training and with total movement control, even under vibrating conditions. However, this might not happen in less experienced populations, where motion control should be guaranteed. Other protocols used accelerometers for this purpose (*Buscà et al., 2020*).

## CONCLUSIONS

Superimposing vibration seems to be a proper strategy to enhance muscle activity in suspended push-ups. 25 Hz and 40 Hz frequencies provoked similar effects on global activity, and all the muscles analyzed, except in triceps brachii and anterior deltoid in the eccentric phase. Nevertheless, no differences were found between the two frequencies, except in the right external oblique. Vibration also led to a higher value of subjective perception of exertion (OMNI-Res), but no differences were found between the two tested frequencies.

## ACKNOWLEDGEMENTS

The authors thank all the study participants for their time and effort. The authors also thank Mr. Abel Folk for his collaboration in data acquisition.

### Funding

This research was supported by the Secretariat of University and Research of the Ministry of Business and Knowledge of the Government of Catalonia and the European Social Fund grant number 2020 FI_B2 00126 and Obra Social "la Caixa" grant number URL/R26/2019. The funders had no role in study design, data collection and analysis, decision to publish, or preparation of the manuscript.

## Grant Disclosures

The following grant information was disclosed by the authors:
Secretariat of University and Research of the Ministry of Business and Knowledge of the Government of Catalonia.
European Social Fund: 2020 FI_B2 00126.
Obra Social "la Caixa": URL/R26/2019.

## Competing Interests

The authors declare there are no competing interests.

## Author Contributions

- Bernat Buscà conceived and designed the experiments, analyzed the data, authored or reviewed drafts of the article, and approved the final draft.
- Joan Aguilera-Castells performed the experiments, analyzed the data, prepared figures and/or tables, authored or reviewed drafts of the article, and approved the final draft.
- Jordi Arboix-Alió conceived and designed the experiments, performed the experiments, prepared figures and/or tables, and approved the final draft.
- Adrià Miró performed the experiments, prepared figures and/or tables, authored or reviewed drafts of the article, and approved the final draft.
- Azahara Fort-Vanmeerhaeghe conceived and designed the experiments, analyzed the data, authored or reviewed drafts of the article, and approved the final draft.
- Pol Huertas performed the experiments, analyzed the data, prepared figures and/or tables, and approved the final draft.
- Javier Peña conceived and designed the experiments, authored or reviewed drafts of the article, and approved the final draft.

## Human Ethics

The following information was supplied relating to ethical approvals (i.e., approving body and any reference numbers):

The Ethics and Research Committee Board of Blanquerna Faculty of Psychology and Educational and Sport Sciences at Ramon Llull University in Barcelona, Spain (ref. number 1819005D) approved this study.

## Data Availability

The datasets are available at Figshare: Aguilera-Castells, Joan (2022): Suspended push-ups. figshare. Dataset. https://doi.org/10.6084/m9.figshare.20293722.v1.

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
