# Peer review of "Superimposed vibration on suspended push-ups"

_PeerJ, doi:10.7717/peerj.14435_

## Round 0.1 · original submission · Major Revisions

Both reviewers agreed on the merit and interesting approach used in the current research. However, reviewers are also concerned about the clarity of the information provided in the experimental approach and methods. It is strongly recommended that you perform a major revision with the aim of improving the replicability.

·

Basic reporting

The authors of manuscript #75972 “Superimposed vibration on suspended push-ups”, investigated the effects of two popular exercise modalities, suspension training and whole-body vibration (with two vibration frequencies) on upper body muscle EMG activity. They found higher muscle EMG activity when the vibration stimuli were superimposed on the suspension exercise. However, there were no apparent differences between vibration frequencies. In general, the submitted manuscript is well organized and well written.

Experimental design

The investigated research questions are well-defined and meaningful. However, there are several inconsistencies and missing details that should be improved.

The introduction is, in general, well written. The secondary hypothesis of the study was related to the perceived exertion during the exercise. However differences in perceived exertion were not mentioned in the introduction. It would be helpful to the reader to introduce the effects of vibration training on perceived exertion.
Although it is not mandatory, your manuscript will benefit from a figure (namely figure 1) depicting the measurement position, including the vibration device, study design and some sample of the EMG data.

I must express some concerns about the EMG analysis used in this study. From the methodology section is not clear if the EMG data were smoothed and which data was further used in statistical analysis. Using the unsmoothed peak EMG activity could be misleading and biased by synchronization in motor unit (MU) firing. In particular, when vibrations are applied, stretch reflexes cause synchronous MU firings (or at least a lot more synchronously compared to voluntary contraction alone), resulting in summation of the EMG signal, which can be seen as spikes in activity in the EMG raw signal. By taking the peak of this spike as your reference point, you are probably overestimating the effect of muscle activity.
Moreover, how can you be sure that the increment in EMG activity during the exercises with superimposed vibrations is only caused by muscle activity and that the vibrations cause no motion artefacts? There are some divergences in the literature regarding detecting motion artefacts during vibrations. For example, some authors, Ritzmann et al , (2010, 10.1007/s00421-010-1483-x) showed that motion artefacts are negligible during vibrations. But other authors pointed out that motion artefacts are a major problem while measuring EMG activity during vibrations and must be removed (Karacan et. al 2014, 10.3389/fnhum.2014.00536). In the article, you did not address this problem. I understand that artefact detection and removal is not a trivial task and it is probably out of the scope of this paper however, you should at least address this problem and provide some evidence (or reference from literature) why your measurement (without additional data analysis) reflects the physiological response.

I have a comment regarding the statistical analysis. Taken singularly, all the used methods are correct; however, there are some inconsistencies.
First, I suppose you calculated the sample size (power analysis) imputing ANOVA data, since GPower software does not provide the possibility to calculate the power of mixed models. However, you used linear mixed models to perform the statistical analysis. If I am not mistaken, the power calculations for ANOVA cannot be directly used for linear mixed models. I am not saying that the sample size is insufficient, but you should use and report the proper tools. Please provide some more information on how you calculated the a priori power. This article could help (https://www.journalofcognition.org/articles/10.5334/joc.10/)
Second, it is unclear if you adjusted post-hoc tests for multiple comparisons after the mixed model.
Third, I don’t understand the added value of using the magnitude-based inference (MBI) in this study. I appreciate the addition of interpreting the effect sizes to add information on the magnitude of the difference. However, you already found statistical differences between groups using mixed models and post-hoc tests. Why do you need another tool to do a similar analysis? Correct me If I am mistaken (I am not a huge expert in MBI analysis), but the spreadsheet you used is meant for comparing two independent groups, but your study design is composed by repeated measures. In addition, how does this statistical method correct for multiple comparisons since you are comparing SHAM vs 25Hz, SHAM vs 40Hz and 25Hz vs 40Hz? If it does not, don’t you think the results you presented could be an overestimation of the effect? Does the MBI framework offer some other solutions (spreadsheets) to analyze repeated measures data?

Validity of the findings

The discussion is well structured and written; however, I cannot comment on it further until the issues with the EMG analysis are clarified.

Additional comments

Specific comments:
Line 108: change “upper body activation” with “upper body EMG activity” or “upper body muscle activity”.
Line 141: not sure what “descriptive data” means in this context.
Line 169: the concept of lower and upper positions is unclear in this context. Did you mean lowering movement and upward movement? How the eccentric or concentric phase could be described by only one position (lower or upper) is unclear. Please rephrase this and check the whole document, since this classification has been used several times.
Line 172: the phrase “metronome settled at 60bpm” is not clear enough how fast the movement was carried out. Was it the same for the concentric and eccentric phases? Please be more explicit.
Line 188: The sentence starting with “sEMG signal normalization” describes the MVIC procedure. I suppose the normalization has been done using the peak smoothed EMG activity from every MVIC measurement. Please correct this paragraph. As described in the general notes, here I expect some explanation of EMG analysis and how the artefact was removed or why they were not removed. As described before, it is unclear which part of the EMG signal was then analyzed.

Line 216: “The data analysis consisted of recording each…” in this step you analyzed the data.

Line 220: Was this part (classification) further developed in this article? If it is not developed and presented, please remove it.

Line 227: In addition to the general comment raised above, it is unclear why you chose this particular effect size in the a priori power analysis (0.32). Is there any study suggesting this ES, or is it your decision (based on what)? Please clarify

Line 229: You should also check for normality of the residuals since this is the assumption for using the linear mixed models.

Line 231: Not clear what muscle activity is… is the peak EMG amplitude or the mean… Moreover, the list of muscles is different than before.

Line 232: what is the added value of analyzing the global activity? I cannot relate how this global activity has physiological value. Can you further explain the use of such an index?

Line 236: It is not clear that the muscle activity was analyzed separately for the concentric and eccentric phases.

Line 239: It is unclear what p adjustment you used for post-hoc tests.

Line 248: minor inconsistency using the letter “d”. It is present in (d>2.0) but missing in other intervals.

Table 1: The activities of the external oblique muscles are really low (4-5% of maximal activity). I would expect much more activity from trunk stabilizers in a suspension exercise. Can you comment on this?

Figure 1: The upper limit of muscle triceps brachii is so close to the mean ES. Is this correct. Please check the data since we expect upper and lower confidence intervals to be equally distant from the means.

·

Basic reporting

The language is direct and objective. There is no ambiguity or terms that make understanding difficult.
In the introduction, only references were cited corroborating the application of vibration during the exercise. However, I believe it is important to cite studies that did not have a significant effect on vibration (Rønnestad et al., 2012; Drummond et al., 2014). The citation of these studies would facilitate the understanding of knowledge gaps in the area and, consequently, it is easier to justify the present study.
Even if it is not required by the journal, it is recommended to use the same formatting in all references in the manuscript.
The structure of the manuscript, as well as the figures and data, are following the norms of the journal.
The submission matches the Self-contained criterion, containing results relevant to the hypotheses and to the area in general.

Experimental design

The research consistently presents the objectives and scope of the journal.
The technical standard of the survey matches the requirements.
The vibration frequency used was 25Hz and 40Hz. Insurin (2005) reports that lower frequencies propagate better through the tissues, but generate a greater effect on the head, which can be harmful (ABERCROMBY, et al., 2007). Has the security of the application been verified in the volunteers? Lines 108-109.
In lines 160-161 it is explained that, to control the extent of hand separation, a barrier has been placed. However, when doing the exercise leaning on the barrier, don't we lose the characteristic of greater freedom of suspension training? And this same lock does not influence the vibratory impulse?
The variation of the angle of inclination can alter the participation of the muscular groups in the exercise and even alter the intensity of the exercise (Rodríguez-Ridao et al., 2020; Saeterbakken et al., 2017). How might this have influenced the results? Line 166.
In line 176, it was mentioned that the vibration amplitude used was 8 mm. Why was this range chosen? Was it measured in the equipment that provided the vibration or in the displacement imposed on the volunteer?
In the material and methods section, an image would help to understand the positioning of the volunteer and especially the vibration equipment. Did the vibration occur in the same direction as the resultant forces (parallel to the arms)?The description presented on lines 177-179 and 338-339 is not sufficient for full understanding.

Validity of the findings

The manuscript results present a reasonable degree of impact and news.
To what extent can we say that the greater muscle activity was provided by the vibratory stimulus, in which the muscle spindles are activated, increasing activity, or simply because the vibration provided greater instability? In the case of the second hypothesis, would any device used to increase instability provide the same effect of increasing muscle activity? Lines 323-325 and 364-365.
All data were provided and have satisfactory robustness in addition to being statistically solid and controlled.
The conclusions are well formulated, related to the research question and focused on the results found.

Additional comments

Line 281: complete the information. "...clavicular head of..."
Line 373 lacks a period after the word “work”.

---

## Round 0.2 · accepted · Accept

The article has merit, and the main concerns were solved during the revision stage. The report presents quality now, and no significant flaws can be identified. Considering this, I would like to suggest acceptance.

·

Basic reporting

The investigated research questions are well-defined and meaningful.

Experimental design

The experimental design was well-conceived and executed.

Validity of the findings

The authors have satisfactorily addressed my comments and substantially improved the quality of the manuscript. I have no more comments.